# The COP9 Signalosome: A Multi-DUB Complex

**DOI:** 10.3390/biom10071082

**Published:** 2020-07-21

**Authors:** Wolfgang Dubiel, Supattra Chaithongyot, Dawadschargal Dubiel, Michael Naumann

**Affiliations:** 1Institute of Experimental Internal Medicine, Medical Faculty, Otto von Guericke University, Leipziger Str. 44, 39120 Magdeburg, Germany; supattra.chaithongyot@med.ovgu.de (S.C.); ddubiel@med.ovgu.de (D.D.); naumann@med.ovgu.de (M.N.); 2School of Pharmaceutical Sciences, Fujian Provincial Key Laboratory of Innovative Drug Target Research, Xiamen University, Xiang’an South Road, Xiamen 361102, China

**Keywords:** COP9 signalosome, ubiquitylation, cullin-RING ubiquitin ligases, DUBs, USP15, USP48, DEN1

## Abstract

The COP9 signalosome (CSN) is a signaling platform controlling the cellular ubiquitylation status. It determines the activity and remodeling of ~700 cullin-RING ubiquitin ligases (CRLs), which control more than 20% of all ubiquitylation events in cells and thereby influence virtually any cellular pathway. In addition, it is associated with deubiquitylating enzymes (DUBs) protecting CRLs from autoubiquitylation and rescuing ubiquitylated proteins from degradation. The coordination of ubiquitylation and deubiquitylation by the CSN is presumably important for fine-tuning the precise formation of defined ubiquitin chains. Considering its intrinsic DUB activity specific for deneddylation of CRLs and belonging to the JAMM family as well as its associated DUBs, the CSN represents a multi-DUB complex. Two CSN-associated DUBs, the ubiquitin-specific protease 15 (USP15) and USP48 are regulators in the NF-κB signaling pathway. USP15 protects CRL1^β-TrCP^ responsible for IκBα ubiquitylation, whereas USP48 stabilizes the nuclear pool of the NF-κB transcription factor RelA upon TNF stimulation by counteracting CRL2^SOCS1^. Moreover, the CSN controls the neddylation status of cells by its intrinsic DUB activity and by destabilizing the associated deneddylation enzyme 1 (DEN1). Thus, the CSN is a master regulator at the intersection between ubiquitylation and neddylation.

## 1. Introduction

The COP9 signalosome (CSN) is a multiprotein complex representing a hallmark of eukaryotic cells. The CSN was discovered as a repressor of constitutive photomorphogenesis (COP) in *Arabidopsis* [1,2] and first isolated from cauliflower [3,4]. Purification from mammalian cells characterized the complex as signaling particle (signalosome) possessing homology to the 26S proteasome lid [5,6,7]. In *Mammalia*, core CSN is composed of six proteasome-COP9-initiation factor 3 (PCI) and two Mov34-and-Pad1p N-terminal (MPN) domain subunits [8,9], essential for CSN function. The 3.8 Å resolution CSN crystal structure based on human recombinant subunits provided detailed information about the subunit-subunit interactions [10]. The PCI domain proteins oligomerize via their winged-helix subdomains in the order of CSN7-CSN4-CSN2-CSN1-CSN3-CSN8, forming a horseshoe-like structure (Figure 1). The MPN domain heterodimer (CSN5, CSN6) is situated on top of the helical bundle formed by the C-terminal α-helices of each CSN subunit [10]. This architecture is shared by paralog complexes of the CSN: the 26S proteasome lid and the translation initiation factor 3 (eIF3) [10,11].

The CSN is substantially more heterogeneous than indicated by the structure of the eight-core subunits. It is associated with kinases and modified by phosphorylation [18,19,20]. For instance, in response to DNA damage causing double strand breaks the ATM kinase phosphorylates CSN3 [21] and UV damage leads to modification of CSN1 [22] with consequences for DNA damage repair. Thus, phosphorylation/dephosphorylation in response to signaling processes produces a dynamic heterogeneity of CSN complexes. Moreover, it has recently been recognized that a fraction of cellular CSN contains a non-essential, non-canonical component called CSNAP [23,24,25]. A further unexplored source of heterogeneity is provided by the fact that several CSN core subunits occur as paralogs/isoforms [26,27]. CSN subunit isoforms originate from gene duplication [26] or from use of alternative translation start sites as shown for CSN8A and CSN8B [28]. They are integrated into distinct CSN variants, which coexist in cells. In *Arabidopsis*, the CSN variants CSN^CSN5A^ and CSN^CSN5B^ confer different physiological functions [2] and in human cells CSN^CSN7A^ and CSN^CSN7B^ have distinct roles in adipogenic differentiation [26,29].

## 2. The CSN Belongs to the Deubiquitylating Enzymes (DUBs)

The ~100 DUBs encoded by the human genome belong to the following families: the ubiquitin (Ub)-specific protease (USP), Ub-C-terminal hydrolase (UCH), JAB1/MPN+/MOV34 protease (JAMM), ovarian tumor protease (OTU), Josephin [30], the novel motif interacting with Ub-containing DUB (MINDY) [31] and the recently discovered ZUFSP/ZUP1 family [32]. DUBs are pivotal regulators of the Ub system involved in protein turnover, signaling, sorting and trafficking [33]. Whereas most DUBs are cysteine proteases, the CSN and its paralog complex lid are metalloproteases of the JAMM family with conserved His, Asp and Ser coordinating a catalytic Zn^2+^ [30]. CSN5 is the only CSN subunit possessing the JAMM motif. Of note, free CSN5 is inactive [12,34] similar to its paralog subunit RPN11 of the 26S proteasome lid [35]. Furthermore, within the CSN complex, CSN5 is in an auto-inhibited state, the Ins-1 conformation [10]. Active CSN5 specifically removes NEDD8 from isopeptide-bonds with conserved Lys residues of cullins. The deneddylating activity of the CSN is activated by its substrates, neddylated cullin-RING-Ub ligases (NEDD8-CRLs) [36], which change the conformation of the active site called induced fit [37]. Structural data provide evidence for coordinated domain changes of CSN2, CSN4, and CSN7 and the CSN5-CSN6 heterodimer induced by neddylated CRL4A converting CSN5 into its active conformation [37]. Obviously, the CSN complex is an essential platform for CSN5 to act as a DUB. Purified mammalian CSN is a DUB specific for NEDD8 and unable to cleave Ub-AMC [12]. Using polyubiquitylated CUL4A as substrate and mutated CSN5, deubiquitylating activity was detected in crude Flag-CSN pulldowns from HeLa cells demonstrating the existence of CSN associated DUBs. Since wildtype CSN5 exhibited different deubiquitylating activity as compared to mutant CSN5, it was assumed that CSN5 has a deubiquitylating activity on its own [38]. However, the data might just reflect an impact of CSN5 on associated DUBs.

Neddylation [39] and deneddylation constitute a regulatory cycle, in which deneddylation inactivates CRLs [37,40] and NEDD8 conjugation stimulates CRL activity by multiple mechanisms [41,42,43,44]. Furthermore, CSN-mediated deneddylation is a prerequisite for the exchange of hundreds of substrate receptors (SRs) including F-box and BTB-domain SRs [45,46,47] as part of a rapid adaptation to altered protein degradation requirements [48]. In this process the CSN cooperates with Cullin-Associated and Neddylation-Dissociated 1 (CAND1) to accelerate the exchange of SRs to optimize the CRL network in response to fluctuations in substrate availability [45,46,49,50,51,52,53,54].

Recently, a potent and specific inhibitor of CSN-mediated deneddylating activity has been discovered, which is called CSN5i-3 [55]. The compound blocks cullin deneddylation and traps CRLs in the neddylated state. CSN5i-3 affects the viability of many tumor cells and suppresses growth of human xenografts in mice [55]. This excellent tool stimulates current and future research on CSN mechanisms and tumor therapy.

In summary, the CSN is a DUB of the JAMM family, controlling the Ub-dependent protein degradation mediated by CRLs, which is essential for maintaining processes such as cell cycle [56], DNA repair [57] and differentiation [48].

## 3. The CSN and Its Paralog 26S Proteasome Lid Cooperate with Diverse DUBs

Analyses of the CSN isolated from different cells by chromatography [14], pulldowns [29], immunoprecipitation [58] as well as density gradient centrifugation [17] revealed its association with additional DUBs, such as USP15 and USP48 and presumably other DUBs as well as with DEN1/NEDP1/SENP8, a member of the SENP family (Figure 1). Thus, the CSN occurs as a multi-DUB complex. USP15 and USP48 belong to the USPs, the largest DUB family, with more than 50 members in *Mammalia* [30]. Both are characterized by one (USP48) or two (USP15) UBL domains and are involved in multiple unrelated biochemical pathways and cellular responses. USP15 activity has been associated with parkin-mediated mitochondrial ubiquitylation and mitophagy [59] and the nuclear factor erythroid 2-related factor 2 pathway in an anti-oxidant response [60]. USP15 has also been shown to stabilize the CRL component RBX1 [61] as well as adenomatous polyposis coli (APC), a subunit of the β-catenin destruction complex [62] and to regulate transforming growth factor-β signaling [63]. There are less reports on USP48 engagements. USP48 stabilizes TRAF2 influencing the E-cadherin-mediated adherens junctions [64]. Whether all these USP15 and USP48 activities need CSN association is not clear at the moment. In the review we focus on CSN associated USP15 and USP48 and their functions in the NF-κB pathway as well as on DEN1, an associated deneddylase.

RPN11, the paralog to CSN5, is the intrinsic DUB of the lid, which also belongs to the JAMM-DUB family. Similar to CSN5, the Ins-1 loop of RPN11 undergoes conformational transition from inactive to active state, which is, in case of RPN11, directed by Ub and ATP [65]. In analogy to the CSN5-CSN6 heterodimer, RPN11 partners with another MPN domain protein, RPN8, possessing an inactive JAMM domain. The activated lid specifically cleaves Ub chains and promotes protein degradation by the 26S proteasome. A deneddylating activity of the lid was not reported. In the 19S regulatory particle the lid cooperates with USP14, a DUB of the USP family, and UCH37 of the UCH family. USP14 and UCH37 are not integral subunits of the 26S proteasome, they assist the lid in removing ubiquitin from substrates to ensure the function of the proteasome [66]. Interestingly, USP14 and UCH37 bind to RPN1 and to RPN13, respectively, which are, in addition to the RPN10, substrate receptors of the 19S regulatory particle [67], which provide a versatile binding platform for various ubiquitin chains [68]. Thus, a coordinated deubiquitylation of incoming substrates within the 19S regulator is presumably necessary for proper function of the 26S proteasome, which is accomplished by the cooperation of lid, USP14 and UCH37. However, since the lid is not directly associated with USP14 and UCH37, it is just part of a multi-DUB complex within the 19S particle.

Unfortunately, so far just few data are published on a possible intrinsic DUB activity of the other paralog complex, the eIF3 [69] and nothing is known about associated DUBs.

Similar structural principles as in the CSN and the lid are mirrored in the BRCA1-A complex in which the active JAMM domain DUB, BRCC36, interacts with the inactive JAMM protein ABRAXAS, whereas in the BRISC complex BRCC36 is supported by ABRO1 [70]. The main substrates of both complexes are Lys63-chains. Their functions, however, are completely different. Whereas BRCA1-A complex serves in DNA double-strand break repair sequestering BRCA1, the BRISC complex is involved in immune signaling. Thus, in this case complexes confer different targeting and specific regulatory functions to BRCC36 DUB, although the substrate remains the same [70]. In case of CSN and lid, the context of their multi-protein complexes provides substrate specificity to the DUBs as well as specific functions.

## 4. CSN-DUB Interactions and Their Role in NF-κB Regulation

The CSN is a signaling platform and in cooperation with the associated USP15 (Ubp12p in *Schizosaccharomyces pombe* or UspA in *Aspergillus nidulans*) it is involved in the NF-κB pathway. Binding of USP15 to the CSN seems to be conserved. According to studies in *S. pombe* the CSN is necessary for efficient transport of Ubp12p to the nucleus. In absence of CSN5, Ubp12p is entirely lost from the CSN [14]. In *A. nidulans*, UspA presumably interacts with the helical bundle of the CSN (see Figure 1) [15]. In human cells, the association between the CSN and USP15 was demonstrated by diverse methods [61]. However, to date, the exact subunits and the CSN variants involved in USP15 interaction are not known.

The CSN recruits USP15 to protect components of CRLs during the remodeling process of the E3 ligase complexes [47,61]. Recently, UspA-dependent protection of the F-box protein F-box23 was shown in *A. nidulans* [15]. Interestingly, F-box23 is presumably the paralog of β-TrCP in mammalian cells, the SR of CRL1^β-TrCP^ responsible for the ubiquitylation of IκBα [71]. By stabilizing the F-box23, the CSN-associated UspA reduces protein levels of the fungal NF-κB-like velvet domain protein VeA, which coordinates differentiation and secondary metabolism [15]. Further, USP15 deubiquitylates IκBα enhancing its stability. Thus, USP15 plays a critical role in the re-accumulation of IκBα in the cytoplasm after induction of the NF-κB pathway, which contributes to the termination of the NF-κB signal [72,73] (Figure 2). CSN1 has a specific role in this context: the N-terminus of CSN1 contacts the CRL1^β-TrCP^, whereas its C-terminus directly interacts with IκBα [74]. Studies on atherogenesis also highlight the CSN function in NF-κB regulation demonstrating the protection of IκBα from degradation by endothelial CSN5/CSN leading to reduction of NF-κB activation upon tumor necrosis factor (TNF) stimulation [75].

The ubiquitin-specific protease 48 (USP48) was identified as a predominately nuclear CSN-associated DUB by co-immunoprecipitation [58]. It stabilizes the nuclear pool of the NF-κB transcription factor RelA upon TNF stimulation by counteracting the ECS^SOCS1^/CRL2^SOCS1^ E3 ligase activity (Figure 2) [58,76]. The catalytic domain of the deubiquitylase USP48 directs the interaction with the Rel homology domain of RelA [77]. Mechanistically, CSN-associated USP48 trims K48-linked ubiquitin chains of RelA, thus controlling the NF-κB activity.

Its trimming activity is enhanced by CK2-mediated phosphorylation and inactivated by dephosphorylation [58]. CK2 is a highly conserved protein kinase possessing brought substrate specificity with pro-proliferative and anti-apoptotic activities. It is constitutively active and messenger-independent [78]. There is a multitude of mechanisms potentially contributing to regulate CK2 activity including localization, turnover and association with other proteins [79]. Most interestingly, the CSN is associated with CK2 [18], highlighting its role as a signaling platform presumably controlling the activity of associated DUBs. In this context, USP15 also is phosphorylated by CSN-associated CK2 [61]. However, the function of this modification is still obscure.

Collectively, USP15 and USP48 are the only two CSN-associated DUBs characterized so far. Additional work is necessary to understand their exact interplay with distinct CSN variants. In addition, unexplored DUBs interacting with specific CSN variants are challenging for future investigations. For this purpose, additional CSN associated DUBs already identified in mass spectrometry experiments [80] have to be verified by appropriate methods such as chemical crosslinking mass spectrometry or Bio-ID, followed by binding studies and mutational analysis.

## 5. CSN-DEN1: The Interplay of Two Major Deneddylases

The physical interaction between the CSN and DEN1 is conserved from fungi to human [17]. In *A. nidulans*, direct protein-protein interaction was determined predominately between DEN1 and CSNG/CSN7, but, in addition, with CSNA/CSN1, CSNE/CSN5 and CSNF/CSN6. In human cells, DEN1 preferentially binds to the N-terminus of CSN1 but there is a weak interaction with CSN2 as well (Figure 1) [17]. To date, it is unknown whether DEN1 specifically binds to a certain CSN variant. Moreover, the exact function of the CSN-DEN1 interaction is still a matter of debate.

DEN1 is a cysteine protease belonging to the Ub-like protease (ULP) family closely related to DUBs [16,30,81,82]. DEN1 crystal structure (Figure 1) reveals the classical catalytic triad, Cys, His and Asp, and the structural basis for its high specificity for NEDD8 [16]. DEN1 can process C-termini of NEDD8 and deconjugates hyperneddylated cullin 1 (CUL1) [83]. However, in fungi NEDD8, processing seems not be an important function of DEN1. Processed NEDD8 was not sufficient to rescue a *denA* deletion mutant [17]. Moreover, the efficiency of DEN1 to cleave NEDD8-CUL1 substrate in vitro is about 100 times lower as compared to the CSN [12]. Therefore, DEN1 is not just another deconjugase of neddylated cullins but also most likely deneddylates additional proteins modified by NEDD8. In *A. thaliana,* DEN1 removes NEDD8 from a subunit of the NEDD8 activating enzyme (NAE). It is assumed that DEN1 regulates the cellular level of free NEDD8, an important control factor for the neddylation/deneddylation cycle [84]. In *A. nidulans*, DEN1/DENA is necessary for appropriate light reaction. In contrast to the CSN, which is responsible for the fungal sexual development, DEN1 is needed for asexual development. Deletion of *denA* revealed a role for the deneddylase in asexual spore formation during limited pyrimidine supply [17].

In human cells, knockout of DEN1 [85] as well as oxidative stress [86] cause accumulation of NEDD8 conjugates, although neddylation of CRLs might even show a significant decrease [87]. Interestingly, human microvascular endothelial cells lacking DEN1 were unable to neddylate CUL1 and subsequently were not able to activate NF-κB or HIF-1α with consequences for vascular inflammatory response. These studies provide evidence for a role of DEN1 in fine-tuning of the inflammatory response [88].

Recently it was found that under conditions of inhibited DEN1, poly-NEDD8 chains are formed [89]. Under normal conditions, NEDD8 chains are degraded by DEN1. If DEN1 is suppressed as under oxidative stress conditions or by knockout, a specific NEDD8 trimer accumulates, which interacts with poly(ADP-ribose) polymerase 1 (PARP1) and attenuates PARP1 activation. Since hyper-activation of PARP1 results in cell death, accumulation of the tri-NEDD8 prevents this cell fate [89]. Therefore, DEN1 has a pivotal role in the regulation of cell death. Whether this DEN1 activity is dependent on its interaction with the CSN is not clear.

One important function of CSN-DEN1 interaction is the initiation of DEN1 degradation [17]. Interestingly, the stability of DEN1 depends on its phosphorylation status. As shown in fungi, DEN1 degradation is regulated by interaction with the CSN as well as by the interplay of phosphorylation and the phosphatase DipA [90]. Thereby, the CSN balances cellular deneddylase activity, which is essential for fungal development and most likely for prevention of human diseases.

## 6. Concluding Remarks

Recently it became clear that the coupling of Ub conjugating and deconjugating machineries is a common principle contributing to the regulation of complex signaling networks [91]. Ub ligases and DUBs regulate the status of protein ubiquitylation, which is crucial for essential cellular processes such as cell cycle progression, DNA repair, signal transduction, protein quality control and differentiation [30]. The CSN is a signaling platform equipped with kinases that coordinates the action of CRLs and DUBs. The member of the JAMM family reduces ubiquitylation of CRLs via deneddylation, thereby protecting CRL components from autoubiquitylation. To optimize this function, the CSN cooperates with USP15 to stabilize CRLs for reassembly and adaptation to changing cellular requests. Further, CSN-associated USP15 directs deubiquitylation and stabilization of CRL-substrate IκBα, whereas CSN-associated USP48 stabilizes the nuclear pool of RelA, thereby facilitating timely induction and shutoff of NF-κB target genes. In addition, the CSN controls deneddylation activity in cells by determining DEN1 stability.

## Figures and Tables

**Figure 1 biomolecules-10-01082-f001:**
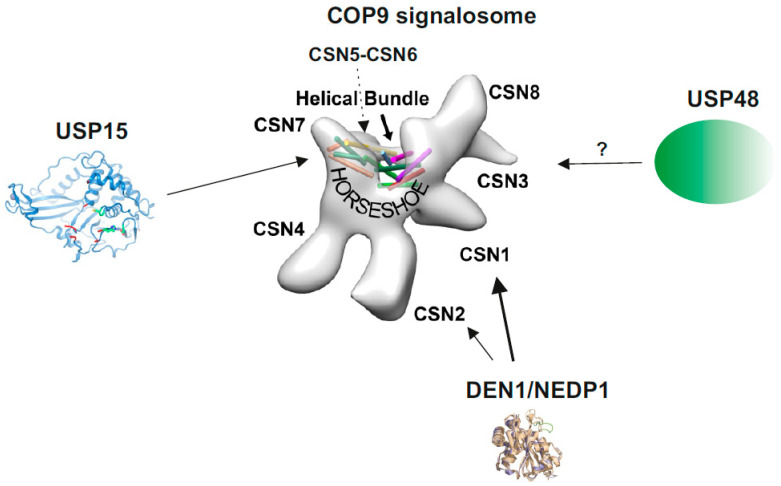
The COP9 signalosome (CSN) and its associated deubiquitylating enzymes (DUBs) and deneddylating enzyme 1 (DEN1). The structure of the CSN was obtained by cryo-electron microscopy using native CSN purified from human red blood cells or from mouse B8 fibroblasts [12]. The localization of CSN subunits, the “Helical Bundle” and the “Horseshoe” structure [10] is indicated. The crystal structure of USP15 is shown with its catalytic core (green) [13]. CSN5 is involved in CSN-USP15/Ubp12 interaction [14]. In addition, in *A. nidulans* USP15/UspA is presumably associated with the entire helical bundle [15]. So far, there is no crystal or cryo-structure of USP48 available. The crystal structure of DEN1/NEDP1 is from Shen et al. [16]. In human cells DEN1 mostly interacts with the N-terminus of CSN1, whereas in fungi it preferentially binds to CSN7 [17].

**Figure 2 biomolecules-10-01082-f002:**
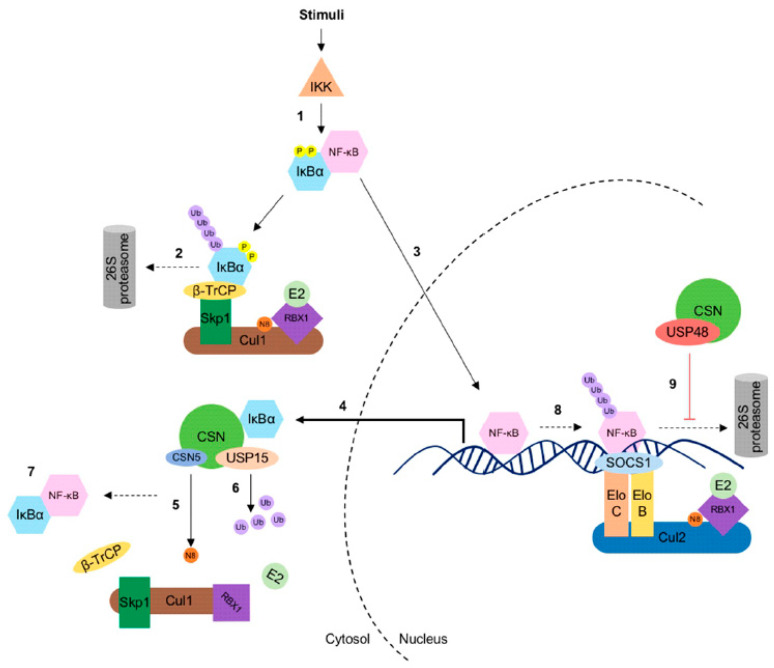
The COP9 signalosome (CSN) partners with ubiquitin-specific protease 15 (USP15) and 48 (USP48) in the NF-κB signaling pathway. (**1**) Upon stimulation, the activated IκB kinase (IKK) complex phosphorylates IκBα, which is subsequently recognized by β-TrCP, the substrate receptor of the CRL1. (**2**) The CRL1^β-TrCP^ ubiquitylates (K48) IκBα, which becomes consequently degraded by the 26S proteasome. (**3**) Due to the degradation of IκBα, NF-κB is released and translocated to the nucleus to activate NF-κB target genes including IκBα. (**4**) De novo synthesized IκBα can be phosphorylated again, (**5**) but the CSN regulates the activity of CRL1^β-TrCP^ by deneddylation and (**6**) CSN-associated USP15, which promotes stabilization and re-accumulation of IκBα, thereby (**7**) terminating NF-κB activation. In a late response at the chromatin, (**8**) the CRL2^SOCS1^/ECS^SOCS1^ targets the nuclear RelA for degradation. However, NF-κB-dependent transcription activity could be sustained by the CSN-associated USP48 activity. (**9**) Herein, USP48 deubiquitylates RelA, which stabilizes RelA at the chromatin.

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
