# Peer review of "The COP9 Signalosome: A Multi-DUB Complex"

_biomolecules, 2020, doi:10.3390/biom10071082_

Round 1
Reviewer 1 Report
The review manuscript by W Dubiel et al. is well written with nice schemas to highlight the Cop9-signalosome complex and associated DUBs. Figure 2 explains very good the role of both DUBs.
The authors have cited extensively the relevant literature. Obviously, this review is refreshing because the perspective of the Cop9 presentation is new. However, the MS does not appear to be sufficiently expanded and therefore lacks background information. The authors could have given a more comprehensive picture, as suggested below:
- I assume that Usp15 and Usp48 belong to the same family of DUBs. Would it be possible to describe this family a bit more? How many members are found in human? what else is known about each of these DUBs? The information in lines 61-62 is too broad.
- The authors report in the introduction that Cop9-s possesses homology to the 26S proteasome lid. Is the 26S proteasome lid also a “Multi-DUB Complex”? Why not write more on clear examples such as Usp14, UCHL5/UCH37 (briefly discussed on lines 97-99). The point of view of this homology discussed by the authors has never been presented before and can make the review very important in the field and highly cited. I look forward to seeing an additional chapter on this topic in the revised manuscript.
- The authors reported very briefly on Cop9-s subunit modifications (citations #12-#14), yet they should have cited two important manuscripts: (1) Meir et al. 1093/nar/gkv270 ; (2) Dubois et al. 10.3892/or.2016.4671 , which showed phosphorylated subunits of Cop9 by the ATM kinase upon both DSB and UV damages. The data is important in order to understand that the Cop9-s is a plastic complex. I would expect the authors to discuss the cop9-s as multiple assemblages - few of them have modified subunits and/or include cop9-s -related DUBs, depending on signaling.
- Lines 147-150: The authors mention CK2 as a csn-interacting protein. This is an important information since CK2 is a kinase that phosphorylates Usp15. Is it possible to provide a comprehensive description of this kinase?
- Lines 109-113: The evolutionary conserved linkage USP15s is described. Please open this paragraph by describing the conservation.
- BTW – Groisman et al. 2003 1016/s0092-8674(03)00316-7 provided some information on a csn-linked DUB.
Author Response
To: Editor
biomolecules
Biomolecules Editorial Office
MDPI, St. Alban-Anlage 66
4052 Basel
Switzerland
Tel: +41 61 683 77 34
Email: biomolecules@mdpi.com
From: Prof. Dr. Wolfgang Dubiel
Institute of Experimental Internal Medicine
Medical Faculty
Otto von Guericke University
Leipziger Str. 44
39120 Magdeburg, Germany
Email: ddubiel@med.ovgu.de
Magdeburg, 3rd July 2020
Re: [Biomolecules] Manuscript ID: biomolecules-841639 - Major Revisions
Review, Title: “The COP9 signalosome: a multi-DUB complex”
Dear Dr. Letty Zhu,
Enclosed please find our revised manuscript titled “The COP9 signalosome: a multi-DUB complex” for publication in biomolecules as a Review.
In your email from 28th June 2020 you have invited us to revise our manuscript taking into account the objections raised by the referees. We have taken detailed account of the helpful comments by the referees as outlined below. Changed and added text is in red.
Yours sincerely,
Wolfgang Dubiel
Point-by-point response to the comments of the reviewers of our manuscript, Manuscript ID: biomolecules-841639
We would like to thank the reviewers for their positive evaluation of our manuscript. In addition, we thank the reviewers for their constructive advices, which were very helpful to improve our manuscript. Specifically we followed the advices of the reviewers and addressed the individual points as follows:
Reviewer 1:
“I assume that Usp15 and Usp48 belong to the same family of DUBs. Would it be possible to describe this family a bit more? How many members are found in human? what else is known about each of these DUBs? The information in lines 61-62 is too broad.”
We are thankful for this advice and added the following information on the USP family and USP15/48 functions in the revised manuscript, page 6, lines 2-12, red text.
“The authors report in the introduction that Cop9-s possesses homology to the 26S proteasome lid. Is the 26S proteasome lid also a “Multi-DUB Complex”? Why not write more on clear examples such as Usp14, UCHL5/UCH37 (briefly discussed on lines 97-99). The point of view of this homology discussed by the authors has never been presented before and can make the review very important in the field and highly cited. I look forward to seeing an additional chapter on this topic in the revised manuscript.”
In a new chapter “The CSN and its paralog 26S proteasome lid cooperate with diverse DUBs”, we compare the CSN with the lid complex. We conclude that the CSN is a multi-DUB complex, whereas the lid is just part of the 19S regulatory multi-DUB complex. For more clarity, the text was rearranged and new text is in red on page 6, lanes 23-26 and page 7, lanes 1-4.
“The authors reported very briefly on Cop9-s subunit modifications (citations #12-#14), yet they should have cited two important manuscripts: (1) Meir et al. 1093/nar/gkv270 ; (2) Dubois et al. 10.3892/or.2016.4671 , which showed phosphorylated subunits of Cop9 by the ATM kinase upon both DSB and UV damages. The data is important in order to understand that the Cop9-s is a plastic complex. I would expect the authors to discuss the cop9-s as multiple assemblages - few of them have modified subunits and/or include cop9-s -related DUBs, depending on signaling.”
We thank the reviewer for this advice and added the very important DNA repair associated phosphorylation of the CSN to the revised manuscript on page 3, lines 18-21, red text.
“Lines 147-150: The authors mention CK2 as a csn-interacting protein. This is an important information since CK2 is a kinase that phosphorylates Usp15. Is it possible to provide a comprehensive description of this kinase?”
As requested by the reviewer we provide a short description of CK2 in the revised manuscript, page 8, lines 25/26, page 9, lines 1-3, red text.
“Lines 109-113: The evolutionary conserved linkage USP15s is described. Please open this paragraph by describing the conservation.”
“Binding of USP15 to the CSN seems to be conserved.” is added to the text on page 7, line 23.
“BTW – Groisman et al. 2003 1016/s0092-8674(03)00316-7 provided some information on a csn-linked DUB.”
The findings by Groisman et al. 2003, are added to the revised manuscript on page 4, lines 25/26 and page 5, lines 1-4, red text.
Attached please find the revised manuscript.
Reviewer 2 Report
Suggestions for Authors for MS titled “The COP9 Signalosome: a Multi-DUB Complex:
Lines 77-83:
“constitute a regulatory circle”. Change ‘circle’ to ‘cycle’.
Additionally, the role of deneddylation in CRL receptor exchange can be further elaborated here by the introduction of CAND1 (Liu et al., Molecular Cell, 2018, 69, 773-786).
Line 122:
“deubiquitylates IkBa, rising its stability.” The word rising can be exchanged with ‘enhancing.’
Line 153:
“Thus, unexplored DUBs perhaps interacting with specific CSN variants are challenging for future investigations.” This concluding line for the paragraph is too vague. Suggest some concrete ways the interactions of CSN-associated factors can be further understood, such as by chemical crosslinking mass spectrometry followed by mutational analysis of residues identified.
Line 159: “week interaction”. Change to “weak”.
Lines 183-184: A role for DEN1 in inflammatory responses can be further elaborated from Ehrentraut et al J Immunnol, 2013, 190, 392-400.
In the reviewer’s opinion, a paragraph devoted to the CSN5 inhibitor (Csn5i-3; Schlierf et al., Nat. Comm, 2016, 7, 13166, and subsequent studies using this inhibitor) will further help clarify the role played by the CSN complex in different tissues and cancers.
Author Response
Point-by-point response to the comments of the reviewers of our manuscript, Manuscript ID: biomolecules-841639
We would like to thank the reviewers for their positive evaluation of our manuscript. In addition, we thank the reviewers for their constructive advices, which were very helpful to improve our manuscript. Specifically we followed the advices of the reviewers and addressed the individual points as follows:
Reviewer 2:
“circle” was changed to “cycle” on page 5, line 5.
“Additionally, the role of deneddylation in CRL receptor exchange can be further elaborated here by the introduction of CAND1 (Liu et al., Molecular Cell, 2018, 69, 773-786).”
We thank the reviewer for this advice and added CAND1 as a regulator of CRL receptor exchange as well as appropriate citations on page 6, lines 8-12, red text.
“rising” was changed to “enhancing” on page 8, line 10.
““Thus, unexplored DUBs perhaps interacting with specific CSN variants are challenging for future investigations.” This concluding line for the paragraph is too vague. Suggest some concrete ways the interactions of CSN-associated factors can be further understood, such as by chemical crosslinking mass spectrometry followed by mutational analysis of residues identified.”
This is a very important advice and few methods are indicated for future research on page 9, lines 8-13, red text.
“week interaction” was changed to “weak” on page 9, line 20.
“Lines 183-184: A role for DEN1 in inflammatory responses can be further elaborated from Ehrentraut et al J Immunnol, 2013, 190, 392-400.”
Thanks to the reviewer an important function of DEN1 was added on page 10, lines 15-18.
“In the reviewer’s opinion, a paragraph devoted to the CSN5 inhibitor (Csn5i-3; Schlierf et al., Nat. Comm, 2016, 7, 13166, and subsequent studies using this inhibitor) will further help clarify the role played by the CSN complex in different tissues and cancers.”
The reviewer is right, the description of CSN5i-3 as an excellent tool for CSN research should be part of the review. Therefore, we added the red text on page 5, lines 13-17.
See attached revised manuscript.
Reviewer 3 Report
In the manuscript titled "COP9 signalosome: a multi-DUB complex" Dubiel W. and colleagues reviewed the role of the COP9 signalosome (CSN) as platform for de-ubiquitinating enzymes (DUBs). Indeed, the analysis of the CSN isolated from different cells has revealed that some DUBs are physically associated with CSN. The review focus on two classical DUBs, USP15 and USP48, and their relative role in regulating the Nf-kB signaling. Furthermore, the authors discuss the interplay between two deneddylases represented by the CSN itself and DEN1, a cysteine protease displaying high specificity for NEDD8. Being most of these interactions highly conserved from fungi to human the authors discuss and compare their roles among different kingdoms ranging from Plantae to Fungi up to mammalian cells.
On the whole, I truly appreciated the review. However, prior publication some very minor issues need to be slightly refined.
1) I do not fully agree that CSN like 26S proteasome is a hallmark of eukaryotic cells otherwise just for fairness one should also mention ribosome too. Hence, just skip the comparison.
2) pag.2 lines 61-64. When the authors list the DUBs families they forgot the rather recently identified ZUP1 that to be thorough it should be mentioned
3) few typos (e.g. pag. 3 line 92 BRISC instead of BRICS).
Author Response
Point-by-point response to the comments of the reviewers of our manuscript, Manuscript ID: biomolecules-841639
We would like to thank the reviewers for their positive evaluation of our manuscript. In addition, we thank the reviewers for their constructive advices, which were very helpful to improve our manuscript. Specifically we followed the advices of the reviewers and addressed the individual points as follows:
Reviewer 3:
“I do not fully agree that CSN like 26S proteasome is a hallmark of eukaryotic cells otherwise just for fairness one should also mention ribosome too. Hence, just skip the comparison.”
We changed the sentence on page 3, lines 2/3 to “The COP9 signalosome (CSN) is a multiprotein complex representing a hallmark of eukaryotic cells.”
“pag.2 lines 61-64. When the authors list the DUBs families they forgot the rather recently identified ZUP1 that to be thorough it should be mentioned”
We thank the reviewer for this important advice and added the ZUP1 family on page 4, lines 10/11.
“few typos (e.g. pag. 3 line 92 BRISC instead of BRICS).”
BRISC was changed to BRICS on page 7, line 9.
See attached revised manuscript.
Round 2
Reviewer 1 Report
Only a minor revision:
Lines 154-159 - BRCC36 is the catalytic subunit for two multi-protein complexes: BRCA1-A and BRISC and not BRICS.
Author Response
This is a very important advice: BRICS was changed to BRISC.